# Referral to the NHS Diabetes Prevention Programme and conversion from nondiabetic hyperglycaemia to type 2 diabetes mellitus in England: A matched cohort analysis

Rathi Ravindrarajah[1], Matt Sutton[1,2], David Reeves[1,2], Sarah Cotterill[1], Emma Mcmanus[1,2], Rachel Meacock[1,2], William Whittaker[1], Claudia Soiland-Reyes[3], Simon Heller[4], Peter Bower[1,2], Evangelos Kontopantelis[2,5]*

1 Division of Population Health, Faculty of Biology, Medicine and Health, University of Manchester, Manchester, United Kingdom, 2 NIHR School for Primary Care Research, Keele, United Kingdom, 3 Research and Innovation Department, Northern Care Alliance NHS Foundation Trust, Salford, United Kingdom, 4 Department of Oncology and Metabolism, University of Sheffield, Sheffield, United Kingdom, 5 Division of Informatics, Imaging, and Data Sciences, University of Manchester, Manchester, United Kingdom

* e.kontopantelis@manchester.ac.uk

**Data Availability Statement:** The data used in this study was obtained via the Clinical Practice

## Abstract

### Background

The NHS Diabetes Prevention Programme (NDPP) is a behaviour change programme for adults who are at risk of developing type 2 diabetes mellitus (T2DM): people with raised blood glucose levels, but not in the diabetic range, diagnosed with nondiabetic hyperglycaemia (NDH). We examined the association between referral to the programme and reducing conversion of NDH to T2DM.

### Methods and findings

Cohort study of patients attending primary care in England using clinical Practice Research Datalink data from 1 April 2016 (NDPP introduction) to 31 March 2020 was used. To minimise confounding, we matched patients referred to the programme in referring practices to patients in nonreferring practices. Patients were matched based on age ($\geq$3 years), sex, and $\geq$365 days of NDH diagnosis. Random-effects parametric survival models evaluated the intervention, controlling for numerous covariates. Our primary analysis was selected a priori: complete case analysis, 1-to-1 practice matching, up to 5 controls sampled with replacement. Various sensitivity analyses were conducted, including multiple imputation approaches. Analysis was adjusted for age (at index date), sex, time from NDH diagnosis to index date, BMI, HbA1c, total serum cholesterol, systolic blood pressure, diastolic blood pressure, prescription of metformin, smoking status, socioeconomic status, a diagnosis of depression, and comorbidities.

A total of 18,470 patients referred to NDPP were matched to 51,331 patients not referred to NDPP in the main analysis. Mean follow-up from referral was 482.0 (SD = 317.3) and

Research Datalink (CPRD). CPRD provides a research service which provides representative, longitudinal real-time anonymized patient electronic health records data from primary care across UK. The licensing agreement between University of Manchester and CPRD, and the data governance of CPRD prevent the sharing or distribution of patient data to other individuals. Hence any requests for access to data from the study should be addressed to cprdenquiries@mhra.gov.uk. All researchers requiring access will require approval of their proposals from CPRD before data release.

**Funding:** MS, DR, SC, RM, WW, SH, PB and EK disclose NIHR funding through the NIHR HS&DR scheme. Award number 16/48/07, Evaluating the NHS Diabetes Prevention Programme (NHS DPP): the DIPLOMA research programme (Diabetes Prevention Long term Multimethod Assessment). The funders had no role in study design, data collection and analysis, decision to publish, or preparation of the manuscript.

**Competing interests:** The authors have declared that no competing interests exist.

**Abbreviations:** CCI, Charlson Comorbidity Index; CPRD, Clinical Practice Datalink; ISAC, Independent Scientific Advisory Committee; MHRA, Medicine and Healthcare Regulatory Agency; NDH, nondiabetic hyperglycaemia; NDPP, NHS Diabetes Prevention Programme; NHS, National Health Services; NIHR, National Institute for Health Research; RCT, randomised controlled trial; T2DM, type 2 diabetes mellitus.

472.4 (SD = 309.1) days, for referred to NDPP and not referred to NDPP, respectively. Baseline characteristics in the 2 groups were similar, except referred to NDPP were more likely to have higher BMI and be ever-smokers. The adjusted HR for referred to NDPP, compared to not referred to NDPP, was 0.80 (95% CI: 0.73 to 0.87) ($p < 0.001$). The probability of not converting to T2DM at 36 months since referral was 87.3% (95% CI: 86.5% to 88.2%) for referred to NDPP and 84.6% (95% CI: 83.9% to 85.4%) for not referred to NDPP. Associations were broadly consistent in the sensitivity analyses, but often smaller in magnitude. As this is an observational study, we cannot conclusively address causality. Other limitations include the inclusion of controls from the other 3 UK countries, data not allowing the evaluation of the association between attendance (rather than referral) and conversion.

## Conclusions

The NDPP was associated with reduced conversion rates from NDH to T2DM. Although we observed smaller associations with risk reduction, compared to what has been observed in RCTs, this is unsurprising since we examined the impact of referral, rather than attendance or completion of the intervention.

## Author summary

### Why was this study done?

- People with "prediabetes" (nondiabetic hyperglycaemia or NDH) are at high risk of developing type 2 diabetes mellitus (T2DM).

- In England, the NHS Healthier You Diabetes Prevention Programme is offered to adults with NDH, offering lifestyle advice to help reduce people's risk of developing T2DM.

- We examined whether people who were referred to the programme were less likely to develop T2DM, compared to those who were not referred to the programme.

### What did the researchers do and find?

- Across 2,209 practices for which we had data, over 700,000 people were identified with prediabetes and around 100,000 had a code in their health record indicating that they were referred to the programme.

- Multivariable survival analyses, controlled for the characteristics of the participants, indicated that the risk was 20% lower in people referred to the programme.

- In other words, assuming 1,000 referred to NDPP and 1,000 not referred to NDPP, by 36 months since referral, we would expect 127 conversions to T2DM in referred to the intervention and 154 in the group not referred.

**What do these findings mean?**

- Our findings support the decision of the rapid large-scale implementation of the programme in England, rather than a slower or regional introduction, but also support the continuation of the programme and the introduction of similar programmes to the rest of the United Kingdom (i.e., Northern Ireland, Scotland, and Wales).

- Future work should focus on longer-term outcomes, the distinction between delaying or preventing progression to T2DM, and whether certain population groups benefit more from the programme.

- Major limitations to our work include the observational nature of the study, which does not allow us to establish causality, and our focus on the evaluation of referral rather than attendance or completion.

## Introduction

Type 2 diabetes mellitus (T2DM) is a major public health concern that has been rising globally, with over 3 million people in the United Kingdom currently diagnosed with T2DM. T2DM is an impairment in the way the body controls and regulates blood sugar levels, and it is usually developed due to a genetic predisposition to the condition as well as behavioural and environmental factors. Previous studies have shown that both lifestyle modifications through diet and physical activity and medication can prevent progression to T2DM [1]. Nondiabetic hyperglycaemia (NDH or "prediabetes") is a condition with increased blood glucose levels but not in the range to be diagnosed as having T2DM. Although the definition of NDH has changed over time [2], diagnosis of NDH has been consistently associated with increased risk of developing T2DM and of developing other diabetes-related conditions [3].

Thus, the NDH population has been seen as an important group to target, in T2DM prevention.

Lifestyle interventions have also been shown to be as effective as pharmaceutical interventions, if not more, in several large randomised controlled trials (RCTs) in various settings and countries, by facilitating weight loss through diet adjustments and increasing exercise activity. The Da Qing IGT and Diabetes Study was the first large RCT to evaluate a lifestyle intervention, with the cumulative incidence of T2DM at 6 years being 68% in the control group compared to 40% in a diet-plus-exercise group, or a 42% risk reduction [4]. The Finnish Diabetes Prevention Study was one of the first large-scale RCTs to evaluate an intensive lifestyle intervention, finding that cumulative incidence of T2DM after 4 years was 11% (95% CI: 6, 15) in the intervention group and 23% (95% CI: 17, 29) in the control group, with an overall reduction risk during the trial of 58% in the intervention group [5]. The US Diabetes Prevention Program also achieved a 58% relative risk reduction through lifestyle measures (individuals coached one-to-one and followed up intensely by telephone), compared with standard advice [6]. The Indian Diabetes Prevention Programme further confirmed the effectiveness lifestyle interventions to prevent progression to T2DM, although the effect size was smaller, with an observed relative risk reduction of 28.5% for the lifestyle modification group, compared to standard care controls [7]. The evidence on effectiveness has been summarised in 2 large meta-analyses, one in 2007 that showed a risk reduction of 51% [8] and a more recent one (2019) with a pooled overall risk reduction of 53% [9]. This overwhelming body of evidence

has driven large-scale interventions through national or regional Diabetes Prevention Programmes, like the Finnish national DPP "FIN-D2D" [10], and the Victoria-Australia "Life!" programme [11].

From a pharmacological point of view, metformin has been a widely studied medication in T2DM prevention, and it has been found to reduce risk by 31%, and being particularly effective for those who were more obese, had higher HbA1c levels, or were younger [6]. Although metformin is the main pharmaceutical intervention, it has also been shown that glitazones, acarbose, or orlistat can also prevent T2DM [12–14].

In the UK, the National Health Services (NHS) Diabetes Prevention Programme (NDPP) is a behavioural intervention programme led by a partnership of NHS England, Public Health England, and Diabetes UK. The programme was primarily offered through primary care practices (99%) to NDH diagnosed adults aged 18 years and over. Identified individuals were either offered referral while in consultation or sent letters through which they could self-refer. The intervention was carried out by 5 commercial providers, and a phased rollout of the programme was introduced across England in 3 waves, from 2016 to 2019 and full population coverage was obtained by mid-2018 [15]. Informed consent was needed for practitioners to make a referral prior to passing the details to a DPP provider. Participants in the programme had an initial assessment followed by regular group education on nutrition and exercise for at least 16 hours over a period of 9 to 12 months. Data from December 2018 showed that 324,699 individuals had been referred of which 53% attended the initial assessment. Approximately 32,665 individuals had at least one intervention session of which 53% completed [16]. Although the NDPP is based on a strong international evidence base [17], justifying the commissioning of such a large and complex programme requires rigorous evidence that the programme is achieving benefits beyond those delivered by current prevention services.

The rollout of the programme makes formal randomised evaluation problematic. Most cases of T2DM are managed through primary care, and primary care administrative data are increasingly used to study diseases and their management [18]. We use primary care administrative data and a range of complex statistical techniques to provide an estimate of the impact of the DPP in reducing conversion of NDH to T2DM (incidence) and reducing the overall numbers of cases of diabetes.

## Methods

### Data source

UK healthcare is free, publicly funded via the NHS, and over 98% of the population is registered to general practices. We used electronic health records from patients in general practice records from the Clinical Practice Research Datalink (CPRD), collecting detailed and anonymised electronic health records from UK general practices using the VISION and EMIS clinical computer systems, collated in 2 separate databases, CPRD GOLD and CPRD AURUM, respectively. We combined data from these databases and analysed them as a single dataset to maximise sample size in the practice-matching designs.

CPRD GOLD captures approximately 7% of the total UK population, whereas AURUM currently covers 13% of the UK population. Data are mostly recorded by GP staff using version 2 Read codes, a semi-hierarchical clinical classification system containing over 100,000 clinical terms that record a patient's details. Additional patient-level information from secondary care, disease registries and death registration records [19,20], can be linked for approximately 60% of GOLD practices and all AURUM practices. The study period was from 1 April 2016 (the start of the DPP's phased rollout) to 31 March 2020. This study is reported as per the

Strengthening the Reporting of Observational Studies in Epidemiology (STROBE) guideline (S1 STROBE Checklist).

## Cohort, exposure to the DPP, and outcome

The study had a proposed analysis plan, which is provided in S1 Analysis Plan.

Participants with NDH were identified using Read codes, which were actively used by GPs in the study: 44v2.00 (Glucose Tolerance Test impaired), C11y200 (Impaired glucose tolerance), C11y300 (Impaired fasting glycaemia), C11y500 (Prediabetes), C317.00 (Nondiabetic Hyperglycaemia), R102.00 ([D] Glucose Tolerance Test abnormal), R102.11 ([D] Prediabetes), R102.12 ([D] Impaired glucose tolerance test), R10D000 ([D] Impaired fasting glycaemia), R10D011 ([D] Impaired fasting glucose), R10E.00 ([D] Impaired glucose tolerance). Our previous work explored codes used to identify individuals, in the context of the changing definition of NDH as well as the conversion of NDH to T2DM in the population prior to the NDPP rollout [2].

The cohort comprised of people diagnosed with NDH during the study period, with the prevalence reported in Table A in S1 Appendix. We extracted data from a total of 2,209 practices (GOLD: 723; AURUM: 1,486). At least 1 referral was recorded in 1,359 practices (GOLD: 64; AURUM: 1,295). Participants were considered as referred to the programme if they were associated with one of the following codes: 679m000/EMISNQDI236-NHS DPP not completed, 679m100 NHS DPP completed, 679m200 NHS DPP started, 679m400/EMISNQRE591 Referral to NHS DPP, EMISNQAT50-NHS DPP attended, EMISNQDI236-NHS DPP not attended.

The outcome of interest was conversion of NDH to T2DM, during the study period. Individuals diagnosed with T2DM following the NDH diagnosis were considered to have progressed to T2DM during the study period. Patients with a previous record of type 1 diabetes were excluded. Code lists were uploaded to the clinical code lists website https://clinicalcodes.rss.mhs.man.ac.uk/medcodes/article/194/ [21] and are also provided in S1 Codelist.

## Matching

We explored different matching approaches to compare conversion rates from NDH to T2DM, between people who were referred to the NDPP and those who were not referred to the NDPP. Although participation to the programme was very high, at the practice level, it was introduced in waves and that allowed us to use a between-practice matching approach to reduce the risk of unmeasured confounding in referrals within a practice. We matched referring practices to nonreferring practices over the study period, before matching referred people (from the referring practice) to nonreferred people (from the matched nonreferring practice). We defined practices with none or one referral as nonreferring, and those with 20 or more referrals as referring, excluding other practices. Our main analysis used a 1-to-1 propensity score matching approach for practices, nearest neighbour with no replacement. We included practices from the UK, not just England, since our nonreferring practice pool drew heavily from non-English practices, where a similar intervention was not implemented during the study period. However, we conducted sensitivity analyses, with English-only practices (all participating in the programme either when contributing data to our study or later). The variables included in the model were the NDH registers (NDH registers are the number of patients who were identified as having NDH) of each practice, for 2016, 2017, 2018, and 2019, to ensure practices of similar counts in terms of the population of interest were matched (Fig A in S1 Appendix).

The next step involved matching referred people in a referring practice to nonreferred people in the matched nonreferring practice. We did this with replacement, to increase the sample size. This step is widely used in analyses of these databases, and the aim is to refine the selection process and reduce the population pool to relevant controls. To achieve this, exact matching is commonly used on certain key and complete covariates, usually age, sex, and general practice [22–26]. This approach was used to match 1 case up to 5 controls, to increase power, using age (within 3 years), sex, and date of NDH diagnosis (within 365 days). Age and sex are used as standard in such matching approaches, since they are complete variables that are likely to be linked to effect heterogeneity. We also included date of NDH diagnosis, since in previous work we identified a decreasing trend in conversion rates over time [2], and we considered it important to ensure that start dates and lengths of subsequent exposure to NDH were balanced between groups. Following this, we controlled for all other relevant recorded covariates in multivariable analyses (for example, biological parameters), with the cohort size allowing for numerous covariates to be included.

We also carried out various sensitivity analyses on the matching approach, with combinations of different strategies for between-practice, and within-practice matching. All these analyses were replicated in a multiple imputation framework where additional covariates with high levels of missingness were included, like HbA1c levels. Full details of all these alternative approaches are provided in Tables B-E and Figs B and C in S1 Appendix.

## Covariates

Index date was defined as the referral date for referred to NDPP and the matched controls. We extracted information on the covariates relevant to NDH and T2DM: age (at index date), sex, time from NDH diagnosis to index date, BMI, HbA1c, total serum cholesterol, systolic blood pressure, diastolic blood pressure, prescription of metformin, smoking status, socioeconomic status (location based, for practice and patients), and a diagnosis of depression. We also quantified the multimorbidity burden, using the Charlson Comorbidity Index (CCI), excluding diabetes with complications [27,28]. Biological parameters were categorised and included a "missing" category to allow a complete case analysis with no records dropped. Where levels of missingness were considered to be high for a biological parameter (above 50%), it was not included in the main analyses (HbA1c). Socioeconomic information was not included in the complete case analysis due it not being available for patients registered in non-English practices. In the multiple imputation analyses, all covariates were included irrespective of levels of missingness. More information on extraction of the covariates relative to the index date and categorisation, where relevant, is provided in S1 Appendix.

## Statistical analysis

We describe the characteristics of the matched cohort. A parametric survival model with a Weibull survival distribution and shared frailty for practice (random effects) was employed to examine associations between the covariates previously described and conversion to T2DM. Standard errors were obtained through 1,000 bootstrap replications of matched clusters. In addition, custom bootstraps of 1,000 replications were used to obtain average survival curves (and their variability) for referred and nonreferred patients, focusing on 12, 24, 30, and 36 months from referral. These estimates were used to quantify the associations of the intervention into numbers of prevented conversions to T2DM at a particular time point, per 1,000 referrals to the programme. We also used a parametric survival command (stpm2) to obtain average survival curves for the analysed individuals within each group, rather than survival curves for the average person [29]. Stata v16 was used for all analyses.

## Sensitivity analyses

Numerous sensitivity analyses were employed, and combinations of them, on top of the between-practice matching sensitivity analyses:

- 1-to-N practice matching (rather than 1-to-1); 1-to-1 patient matching (rather than 1-to-N) for potentially better balance.

- Sampling controls without replacement.

- Chained equations multiple imputation (10 datasets), with the model including all covariates previously described (and HbA1c, practice and patient location deprivation) as well as the outcome, with continuous variables not categorised.

- Within-practice matching.

- Including practice region and database (GOLD/Aurum) to account for potential regional and recording variation.

- A large group of individuals who were referred to the programme did not have a NDH code or had one after referral were excluded in the main analysis. We carried out additional sensitive analyses including this group of people including their referral date as their NDH diagnosis date.

- An additional sensitivity analysis involved obtaining bespoke linked data from the CPRD to exclude practices where there was a discrepancy in referred to NDPP in their records and the MDS (>20 referrals in the MDS), which could have introduced bias by allowing referred to NDPP to be selected as not referred to NDPP.

## Patient and public involvement

There was no direct patient involvement in the context of this evaluation, since stakeholders' perceptions and experiences of the programme have been evaluated elsewhere [30].

## Ethical approval

This study is based on data from the CPRD obtained under license from the Medicines and Healthcare Products Regulatory Agency (MHRA). The data are provided by patients and collected by the NHS as part of their care and support. The interpretation and conclusions contained in this study are those of the authors alone, and not necessarily those of the MHRA, the National Institute for Health Research (NIHR), NHS, or the Department of Health. Approval to conduct this study using the CPRD was granted by the Independent Scientific Advisory Committee (ISAC) of the MHRA (protocol 18_101). We thank the contributing patients and practices to the CPRD who have allowed their data to be used for research purposes.

# Results

The final cohort, with 1-to-1 matching of practices and individual patients matched up to 5 controls with replacement, included 69,801 participants from 1,084 practices: 18,470 referred to NDPP and 51,331 not referred to NDPP (diagnosed with NDH but not referred to the programme) (Fig 1). Baseline characteristics are shown in Table 1. The mean age of referred to NDPP and not referred to NDPP were fairly similar with referred to NDPP being 61.9 (SD = 11.6) years and not referred to NDPP being 62.6 (SD = 11.0). Although the mean BMI of the referred to NDPP and not referred to NDPP were similar, at 30.8 (SD = 6.4) and 31.2

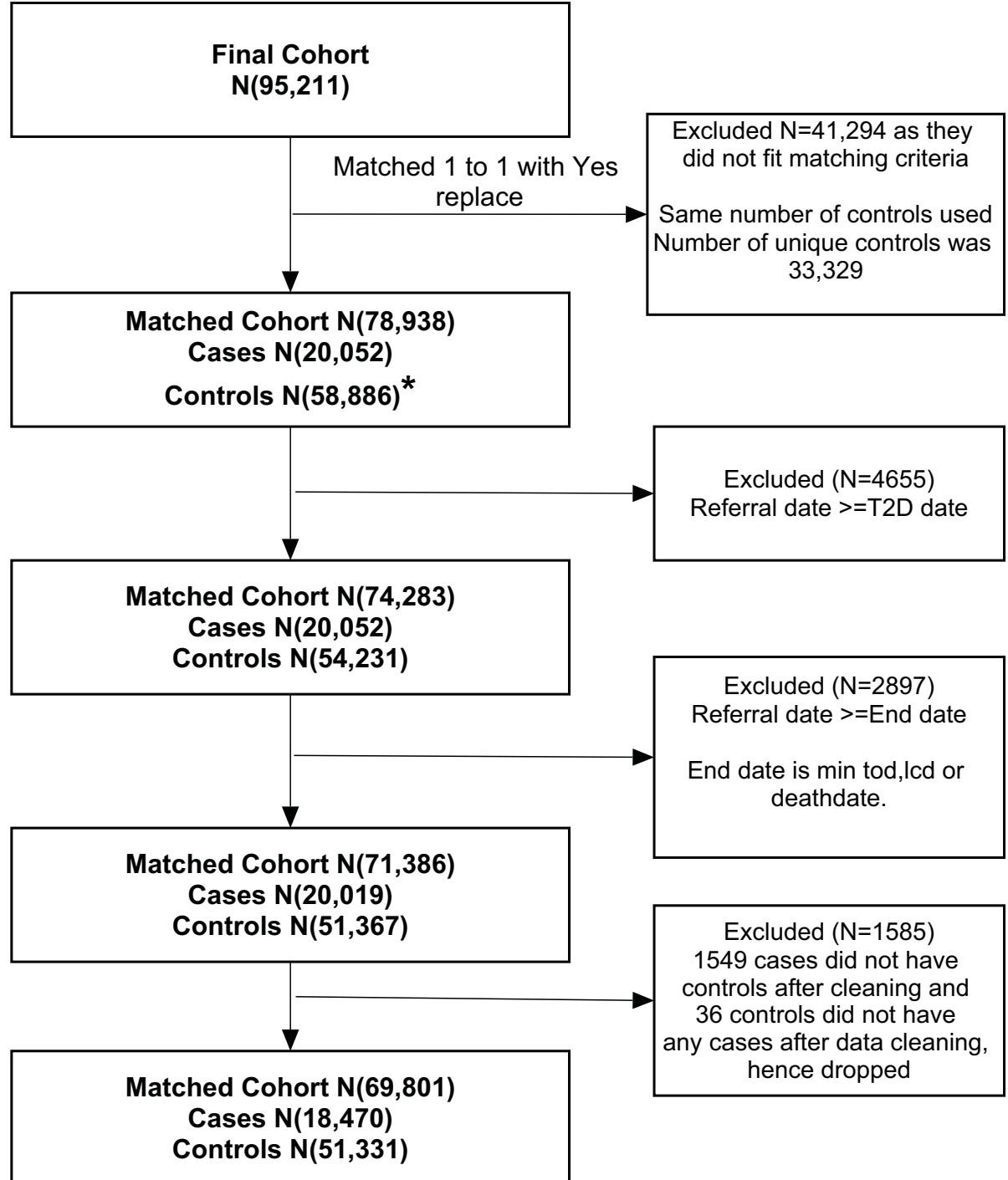

**Fig 1. Flowchart on final matching numbers CPRD AURUM and GOLD.** [Matching: Patients referred to the scheme versus matched patients not referred within the same practice, based on age (up to 3 years), sex, and within 365 days of NDH diagnosis: Cases- Referred to NDPP; Controls- Not Referred to NDPP].

(SD = 6.7), respectively, those who were referred were more likely to be obese, with 34% of referred to NDPP and 29% of not referred to NDPP with a measurement of BMI equal to or above 30 kg/m$^2$. Results showed that 27% of the cohort had depression. Around 27% of the

**Table 1. Characteristics of the matched cohort, N (%) or mean ± standard deviation.** [Cases are those referred to the NHS DPP and Controls are those with NDH diagnosis in primary care, matched for sex and (within 3 years) age within 365 days of NDH diagnosis date].

| | All | Referred to NDPP | Not Referred to NDPP | Standardised Difference |
|---|---|---|---|---|
| **All** | 69,801 | 18,470 | 51,331 | |
| **Follow-up from NDH diagnosis (days)** | 879.9 ± 804.8 | 980.6 ± 902.4 | 843.6 ± 763.4 | 0.17071 |
| **Follow-up from index date (days)** | 474.9 ± 311.3 | 482.0 ± 317.3 | 472.4 ± 309.1 | 0.03084 |
| **Time from NDH diagnosis to Referral date (days)** | 404.9 ± 741.7 | 498.7 ± 835.4 | 371.2 ± 701.9 | 0.17240 |
| **T2DM** | 4,432 (6.3) | 1,152 (6.2) | 3,280 (6.4) | −0.00630 |
| **≥1 year follow-up or earlier T2DM conversion** | 24,393 (62.0) | 6,318 (60.4) | 18,075 (62.7) | −0.02110 |
| **≥2 years follow-up or earlier T2DM conversion** | 11,815 (30.1) | 3,229 (30.9) | 8,586 (29.8) | 0.02020 |
| **≥3 years follow-up or earlier T2DM conversion** | 3,111 (7.9) | 919 (8.8) | 2,192 (7.6) | 0.03420 |
| **Males** | 33,716 (48.3) | 8,941 (48.4) | 24,775 (48.3) | 0.00290 |
| **Females** | 36,085 (51.7) | 9,529 (51.6) | 26,556 (51.7) | |
| **Age (years) NDH diagnosis date** | 62.4 ± 11.2 | 61.9 ± 11.6 | 62.6 ± 11.0 | −0.06271 |
| **Age (years) Referral date** | 63.6 ± 11.3 | 63.3 ± 11.8 | 63.7 ± 11.1 | −0.03543 |
| **Age group (Years) NDH diagnosis date** | | | | |
| **18–34** | 513 (0.7) | 191 (1.0) | 322 (0.6) | −0.05940 |
| **35–44** | 3,690 (5.3) | 1,207 (6.5) | 2,483 (4.8) | |
| **45–54** | 12,934 (18.5) | 3,537 (19.2) | 9,397 (18.3) | |
| **55–64** | 21,601 (31.0) | 5,510 (29.8) | 16,091 (31.4) | |
| **65–74** | 20,828 (29.8) | 5,319 (28.8) | 15,509 (30.2) | |
| **75–84** | 9,252 (13.3) | 2,439 (13.2) | 6,813 (13.3) | |
| **> = 85** | 983 (1.4) | 267 (1.5) | 716 (1.4) | |
| **Body Mass Index (kg/m²)** | 31.1 ± 6.6 | 30.8 ± 6.4 | 31.2 ± 6.7 | −0.06041 |
| **Body Mass Index Categories** | | | | |
| **<18.5** | 325 (0.47) | 85 (0.46) | 240 (0.47) | −0.20880 |
| **18.5–25** | 6,328 (9.1) | 1,973 (10.7) | 4,355 (8.5) | |
| **25–30** | 13,731 (19.7) | 4,300 (23.3) | 9,431 (18.4) | |
| **> = 30** | 21,355 (30.6) | 6,255 (33.9) | 15,100 (29.4) | |
| **Missing** | 28,062 (40.2) | 5,857 (31.7) | 22,205 (43.3) | |
| **Charlson comorbidity score** | | | | |
| **None** | 38,371 (55.0) | 10,362 (56.1) | 28,009 (54.6) | 0.04630 |
| **1 to 2** | 19,109 (27.4) | 4,494 (24.3) | 14,615 (28.5) | |
| **3 to 4** | 7,168 (10.3) | 1,813 (9.8) | 5,355 (10.4) | |
| **>4** | 5,153 (7.4) | 1,801 (9.8) | 3,352 (6.5) | |
| **Cholesterol (%)** | | | | |
| **<3** | 1,182 (1.7) | 352 (1.9) | 830 (1.6) | 0.34690 |
| **3 to 4** | 8,680 (12.4) | 2,603 (14.1) | 6,077 (11.8) | |
| **4 to 5** | 15,905 (22.8) | 4,970 (26.9) | 10,935 (21.3) | |
| **5 to 6** | 13,345 (19.1) | 4,441 (24.0) | 8,904 (17.4) | |
| **> = 6** | 8,118 (11.6) | 2,751 (14.9) | 5,367 (10.5) | |
| **Missing** | 22,571 (32.3) | 3,353 (18.2) | 19,218 (37.4) | |
| **Depression** | 18,893 (27.1) | 4,275 (23.2) | 14,618 (28.5) | −0.12020 |
| **Smoking Status** | | | | |
| **Current Smoker** | 25,751 (36.9) | 9,108 (49.3) | 16,643 (32.4) | −0.44060 |
| **Ex-smoker** | 30,207 (43.3) | 8,029 (43.5) | 22,178 (43.2) | |
| **Never smoker** | 12,697 (18.2) | 928 (5.0) | 11,769 (22.9) | |
| **Missing** | 1,146 (1.6) | 405 (2.2) | 741 (1.4) | |
| **Metformin** | 2,059 (3.0) | 621 (3.4) | 1,438 (2.8) | 0.03310 |

*(Continued)*

**Table 1.** (Continued)

| | All | Referred to NDPP | Not Referred to NDPP | Standardised Difference |
|---|---|---|---|---|
| **Systolic Blood Pressure (mm Hg)** | 135.4 ± 14.4 | 134.9 ± 14.1 | 135.6 ± 14.5 | −0.04863 |
| **<120 mm Hg** | 6,824 (9.8) | 2,035 (11.0) | 4,789 (9.3) | −0.23600 |
| **{120–139} mm Hg** | 28,387 (40.7) | 8,414 (45.6) | 19,973 (38.9) | |
| **{140–159} mm Hg** | 18,440 (26.4) | 5,075 (27.5) | 13,365 (26.0) | |
| **> = 160 mm Hg** | 2,796 (4.0) | 719 (3.9) | 2,077 (4.1) | |
| **missing** | 13,354 (19.1) | 2,227 (12.1) | 11,127 (21.7) | |
| **Diastolic Blood Pressure (mm Hg)** | 79.2 ± 9.2 | 79.4 ± 9.1 | 79.2 ± 9.2 | 0.02180 |
| **<80 mm Hg** | 28,178 (40.4) | 8,139 (44.1) | 20,039 (39.0) | −0.20820 |
| **{80–89} mm Hg** | 21,457 (30.7) | 6,168 (33.4) | 15,289 (29.8) | |
| **> = 90 mm Hg** | 6,812 (9.8) | 1,936 (10.5) | 4,876 (9.5) | |
| **missing** | 13,354 (19.1) | 2,227 (12.1) | 11,127 (21.7) | |
| **HbA1c (mmol/mol)** | 43.4 ± 3.7 | 43.5 ± 2.3 | 43.4 ± 3.9 | 0.02819 |
| **% with HbA1c values** | 22,806 (32.7) | 3,590 (19.4) | 19,216 (37.4) | −0.38940 |

NDH, nondiabetic hyperglycaemia; NDPP, NHS Diabetes Prevention Programme; T2DM, type 2 diabetes mellitus.

cohort had at least one comorbidity according to the CCI. The prevalence of comorbidities according to the CCI score was higher in those referred to NDPP compared to not referred to NDPP, with 10% of those referred to NDPP having more than 4 comorbidities, compared to 7% of not referred to NDPP. Current smokers were more likely to be referred to the programme, with those referred to NDPP having 49% and 32% not referred to NDPP being current smokers. Metformin was prescribed in 3.4% of those referred to NDPP and 2.8% of those not referred to NDPP.

A total of 4,432 (6.4%) participants developed T2DM during the study period, of which 1,152 (6.2%) were referred to NDPP and 3,280 (6.4%) were not referred to NDPP. The mean days from NDH diagnosis to index date was 405 days (SD = 742), and it was higher in those referred to NDPP with a mean of 498 days (SD = 835), compared to 371 days (SD = 702) in those not referred to NDPP. The rate of conversion to T2DM was lower in those referred to NDPP compared to those not referred to NDPP with a HR of 0.80 (95% CI: 0.73, 0.87) (Table 2). Nonconversion ("survival") at 24 months since referral was estimated at 89.9% (95% CI: 89.5% to 90.4%) for not referred to NDPP and 91.8% (95% CI: 91.2% to 92.3%) for referred to NDPP. At 36 months, it was 84.6% for not referred to NDPP (95% CI: 83.9% to 85.4%) and 87.3% for referred to NDPP (95% CI: 86.5% to 88.2), respectively. The difference in conversion rates at 36 months was −2.7% (95% CI: −3.7% to −1.7%). Average survival plots across the 2 groups are presented in Fig 2.

NDH, nondiabetic hyperglycaemia; NDPP, NHS Diabetes Prevention Programme; T2DM, type 2 diabetes mellitus.

Females were less likely to convert to T2DM with a HR of 0.80 (95% CI: 0.75, 0.86) ($p < 0.001$) compared to men. People aged 45 to 54 had a higher risk of conversion to T2DM, compared to those aged 18 to 34, with a HR of 2.13 (95% CI: 1.44, 3.15) ($p < 0.001$). Cholesterol categories did not appear to be strongly associated with conversion to T2DM. People with high BMI had a higher risk of conversion to T2DM, with those classed overweight (BMI 25 to 30) having an HR of 1.54 (95% CI: 1.30, 1.82) ($p < 0.001$), and those classed obese (BMI > = 30) having an HR of 2.34 (95% CI: 2.00, 2.74) ($p < 0.001$), compared to individuals with a normal BMI (18.5 to 25). Having depression at baseline may have increased the risk of conversion (HR = 1.19, 95% CI 1.11, 1.27) ($p < 0.001$). Those who had a prescription for metformin

**Table 2. Cox proportional hazard models exploring time to conversion of referred to T2DM for patients (cases and controls) by baseline characteristics, with shared frailty for practice.**

| | HR (95% CI) | *p*-value |
|---|---|---|
| **NHS DPP cohort** | | |
| **Not Referred to NDPP** | Ref | |
| **Referred to NDPP** | 0.80 (0.73 to 0.87) | <0.001 |
| **Males** | Ref | |
| **Females** | 0.80 (0.75 to 0.86) | <0.001 |
| **Age Group (years)** | | |
| **18–34** | Ref | |
| **35–44** | 1.89 (1.26 to 2.84) | 0.002 |
| **45–54** | 2.13 (1.44 to 3.15) | <0.001 |
| **55–64** | 1.97 (1.33 to 2.92) | 0.001 |
| **65–74** | 1.78 (1.20 to 2.64) | 0.004 |
| **75–84** | 1.66 (1.11 to 2.48) | 0.013 |
| **> = 85** | 1.55 (0.92 to 2.63) | 0.101 |
| **Cholesterol categories (%)** | | |
| **4 to 5** | Ref | |
| **<3** | 1.22 (0.97 to 1.53) | 0.091 |
| **3 to 4** | 1.00 (0.89 to 1.12) | 0.94 |
| **5 to 6** | 0.98 (0.90 to 1.07) | 0.674 |
| **> = 6** | 1.03 (0.92 to 1.15) | 0.606 |
| **Missing** | 0.85 (0.77 to 0.93) | <0.001 |
| **Smoking Status** | | |
| **current smoker** | Ref | |
| **ex-smoker** | 1.00 (0.93 to 1.08) | 0.988 |
| **never smoker** | 1.14 (1.03 to 1.25) | 0.008 |
| **missing** | 1.29 (1.01 to 1.64) | 0.042 |
| **Body Mass Index Categories (kg/m$^2$)** | | |
| **<18.5** | 0.70 (0.05 to 9.92) | 0.795 |
| **18.5–25** | Ref | |
| **25–30** | 1.54 (1.30 to 1.82) | <0.001 |
| **> = 30** | 2.34 (2.00 to 2.74) | <0.001 |
| **Missing** | 1.56 (1.33 to 1.84) | <0.001 |
| **Depression** | 1.19 (1.11 to 1.27) | <0.001 |
| **CCI Score** | | |
| **None** | Ref | |
| **1 to 2** | 1.14 (1.06 to 1.24) | 0.001 |
| **3 to 4** | 1.21 (1.08 to 1.35) | 0.001 |
| **>4** | 1.34 (1.18 to 1.53) | <0.001 |
| **Metformin** | 9.89 (9.03 to 10.82) | <0.001 |
| **Systolic Blood Pressure (mm Hg)** | | |
| **<120 mm Hg** | ref | |
| **{120–139} mm Hg** | 0.82 (0.73 to 0.92) | <0.001 |
| **{140–159} mm Hg** | 0.91 (0.80 to 1.02) | 0.115 |
| **> = 160 mm Hg** | 0.88 (0.73 to 1.06) | 0.181 |
| **missing** | 0.83 (0.73 to 0.95) | 0.005 |
| **Diastolic Blood Pressure (mm Hg)** | | |
| **<80 mm Hg** | | |

(*Continued*)

**Table 2.** (Continued)

|  | HR (95% CI) | *p*-value |
|---|---|---|
| **{80–89} mm Hg** | 1.11 (1.03 to 1.21) | 0.008 |
| **> = 90 mm Hg** | 1.15 (1.02 to 1.31) | 0.026 |
| **missing** |  |  |
| **Days from NDH diagnosis to Referral date** | 1.0002 (1.0002 to 1.0002) | <0.001 |

were at a much higher risk of developing T2DM, with an HR of 9.89 (95% CI: 9.03, 10.82) (*p* < 0.001). Those with a missing smoking status and those who had never smoked were also at a slightly higher risk of developing T2DM, compared to current smokers. Blood pressure was not significantly associated with risk of conversion to T2DM.

## Sensitivity analysis results

Results from the sensitivity analyses are presented in Table D in S1 Appendix, and there was broad agreement with the main analysis, although in most cases the association between referral and nonconversion to T2DM was smaller. The bespoke linkage analyses, where 130 of the 1,084 (12.0%) practices were dropped due to potential data inconsistency, were in agreement with our main analyses. Estimates of associations with risk reduction were lower in within-practice matching analyses, and in between-practice matching analyses that controlled for region.

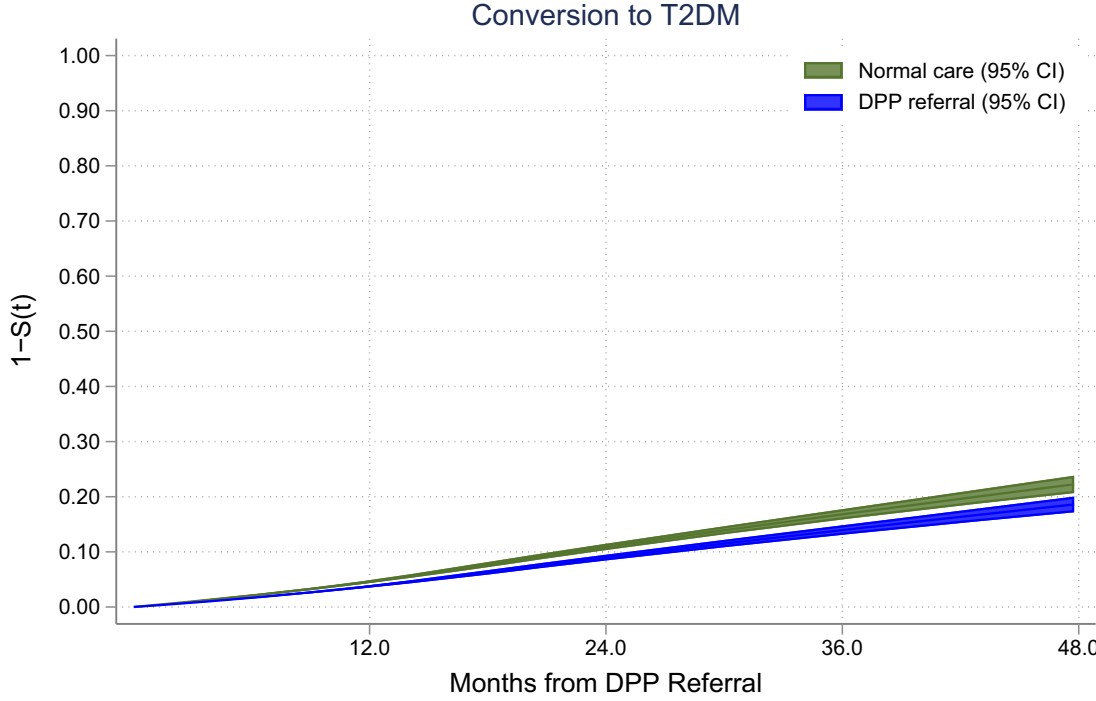

**Fig 2. Conversion of patients referred to the programme and those who received normal care in primary care in the study period.** (Since the flexible parametric model command we used to obtain average survival curves (stpm2) does not allow for random-effects components, which we found to be important in the analyses, we obtained the average of the survival curves within each group for the 1-to-1 patient matched sensitivity analysis. Referrals and nonreferrals were, respectively, 18,470 and 18,470 for t > = 0; 10,466 and 10,444 for t > = 12; 4,148 and 4,122 for t > = 24; 919 and 896 for t > = 36; 0 and 0 for t > = 48. Model adjusted for age (at index date), sex, time from NDH diagnosis to index date, BMI, total serum cholesterol, systolic blood pressure, diastolic blood pressure, prescription of metformin, smoking status, depression, and multimorbidity burden (using the Charlson Comorbidity Index)).

## Discussion

Our findings suggest that individuals who were referred to the programme were less likely to develop T2DM during the study period compared to those who were not referred to the programme. Assuming 1,000 referred to NDPP and 1,000 not referred to NDPP, by 36 months since referral, we would expect 127 (95% CI: 118 to 135) conversions to T2DM in those referred to NDPP and 154 (95%CI: 146 to 161) in those not referred to NDPP. As this is an observational study, we cannot be conclusive on causality.

### Findings in context

Diabetes prevention trials have shown reductions in the incidence of T2DM. Analysis of the DPP provider dataset has shown that individuals who attended the NHS DPP were associated with a significant reduction in weight ("mean weight loss of 2.3 kg [95% CI 2.2, 2.3]") and HbA1c ("an HbA1c reduction of 1.26 mmol/mol (1.20, 1.31) (0.12% [0.11, 0.12]") in an intention-to-treat analysis, which could be a marker of reduction in risk of T2DM [16]. Most of the trial results showed that weight loss was the key factor in reducing risk of T2DM, and our results also suggest that increased BMI to be one of the key factors in developing diabetes.

In our study, metformin prescription was associated with higher risk of developing T2DM, reflecting the fact that metformin is prescribed to people deemed at very high risk. However, our aim was to control our analyses for metformin prescription, not examine its association with developing T2DM. Our data also suggest that metformin prescription was higher in women, which might be related to polycystic syndrome, which carries a higher risk of developing T2DM.

The NHS DPP was primarily based on a systematic review and meta-analysis by Public Health England, which assessed the effectiveness of lifestyle interventions for prevention of T2DM in routine practice over a period of 12 to 18 months. The pooled results from the review showed that those attending a DPP reduced the risk of progression to T2DM by 26% compared to those receiving usual care [17]. These findings could be comparable to our results, which suggested those who were referred to the program had a 20% lower risk of conversion to T2DM compared to those who were not referred, during the study period. However, we were able to evaluate the intervention over a longer period of time, when only a few of the clinical trials included in the systematic review used for the NHS DPP design reported outcomes beyond 12 months. Thus, we were in a better position to explore longer-term benefits, despite the observational nature of the design. A recent 30-year follow-up data from the Da Qing Diabetes Prevention Outcome Study showed that lifestyle intervention in individuals who were at risk of T2DM not only delayed the onset of T2DM but also reduced the incidence of other comorbidities related to T2DM, death, and increased life expectancy [31].

Our risk reduction estimates for conversion to T2DM are considerably lower to what has been reported in the largest RCTs, which reported risk reduction rates between 42% and 58% [4–6]. However, these RCTs recruited people at higher risk, and who can benefit more from an intervention, while the interventions were intensive, with rigorous and persistent follow-up regimes. In addition, the RCTs evaluated participation and completion, while we could only evaluate referral to the programme, irrespective of whether the patient completed or even attended the diabetes prevention programme. One should also bear in mind that there exists variation that we could not account for or model, due to practice anonymity in the CPRD, driven by differences in how the programme was delivered by the 4 contracted different providers to deliver the NDPP, and differences in the characteristics of the people who attended the 4 programmes [15,32,33].

## Strengths and limitations

Our study has several strengths. The data were based on a large, longitudinal sample that is generalizable to the UK population [20,34]. Using CPRD data, we were able to access a complete medical history of the patient including other comorbidities and biological measures. However, several limitations also exist. The matching process involved a lot of complicated decisions. We had to deal with different types of confounding (for example, potential selection bias within a referring practice) and a drop in NDH conversion rates over time (so need to match on NDH diagnosis date). We a priori considered the between-practice matching approach, where we compare referred patients in English referring practices to nonreferred patients in nonreferring English, Scottish, Northern Irish, and Welsh practices, as the least likely to be affected by selection bias. The disadvantage is that the sample size after matching was smaller, and we also were limited in the number of English practices that could serve as controls in the early years (because of the national coverage of the NHS DPP). On the other hand, if 50% of referred to NDPP are not in the GP records, this is the biggest risk with the within-practice approach. However, in analyses where we controlled for region and database type, the observed association was smaller. Another limitation of the dataset is that a large number of individuals referred to NDPP being dropped during the matching process. The restrictions in the matching process were there to protect against unmeasured confounding, as much as possible under these designs. Thus, we accepted dropping individuals referred to NDPP and not referred to NDPP, to ensure that analyses will be more robust. However, further sensitivity analyses were carried out that included referrals without an NDH code, or an NDH code following referral—cases (referred to NDPP) that were excluded from the main analysis. We allowed declined referrals to enter the control (not referred to NDPP) pool (81 unique controls selected 179 times in our main analyses), but excluding them did not affect our results. Our estimates could also be affected by underrecoding of referrals in GP practices and the sensitivity using the bespoke CPRD linkage aimed to examine that. Our results are based on an intention to treat analysis; hence, we were not able to confirm whether all individuals who were referred to the programme actually attended and completed the programme (quality of recording is very poor for these 2 categories). However, as we had a code for those who declined referral, we excluded these individuals from our analysis. Another possible limitation is not being able to include HbA1c in our model due to high levels of data missingness —although we included it in the multiple imputation analyses. HbA1c completeness levels also varied between those referred to the programme and those who did not, with more missing data for those referred, which, on the surface, appears counterintuitive, but that was primarily driven by variation in recording across countries, with better recording in Scotland and Wales. There is also no standard definition for NDH in primary care, the thresholds used to define NDH varied across the years, and this could also have an effect on the cases and controls used in the study [35]. Finally, referred and nonreferred people may have different health-seeking behaviours, with referred people diagnosed with T2DM earlier due to higher levels of interaction with primary care. In that case, we would be underestimating the impact of the intervention.

## Implications for policy and future research

Our findings indicate that the NDPP appears to be successful in reducing progression to T2DM, even when we were only able to examine referral to the programme, rather than attendance or completion, in an observational setting. This supports the decision of the rapid large-scale implementation of the programme in England, rather than a slower or regional introduction, but also supports the continuation of the programme. Our findings also support the

introduction of similar programmes to the rest of the UK (i.e., Northern Ireland, Scotland, and Wales), but we would expect our findings to be generalisable to similarly organised healthcare systems.

Future work should quantify the potential benefits of attending or completing the programme, examine longer-term outcomes, and investigate whether the programme delays or prevents progression to T2DM. In addition, it would be important to examine whether the programme attenuates or increases health inequalities, and also whether certain population groups appear to be benefitting more from the programme (i.e., whether there exists effect/association heterogeneity).

## Conclusions

The NDPP was associated with risk reduction in conversion to T2DM, at least in the short to medium term. Individuals who were referred to the NHS DPP by primary care physicians were less likely to develop T2DM compared to those who received usual care. Although we observed lower levels of associations compared to what has been observed in randomised clinical trials, we examined the impact of referral, rather than attendance or completion of the intervention. It will be important to explore whether the association of the programme varies across population subgroups, as well as the long-term effects of the programme. Finally, further work is needed to examine the long-term effects of the programme and to examine if the programme delays or prevents T2DM.

## Supporting information

**S1 STROBE Checklist. STROBE checklist.**
(PDF)

**S1 Appendix. Additional information on methods and results.**
(DOCX)

**S1 Analysis Plan. Proposed analysis plan.**
(DOCX)

**S1 Codelist. T2DM code list.**
(DOCX)

## Author Contributions

**Conceptualization:** Matt Sutton, David Reeves, Sarah Cotterill, Peter Bower, Evangelos Kontopantelis.

**Data curation:** Emma Mcmanus, Evangelos Kontopantelis.

**Formal analysis:** Rathi Ravindrarajah, Evangelos Kontopantelis.

**Funding acquisition:** Matt Sutton, Sarah Cotterill, Rachel Meacock, William Whittaker, Peter Bower, Evangelos Kontopantelis.

**Investigation:** Rathi Ravindrarajah, Evangelos Kontopantelis.

**Methodology:** David Reeves, Evangelos Kontopantelis.

**Project administration:** Claudia Soiland-Reyes.

**Resources:** Evangelos Kontopantelis.

**Software:** Evangelos Kontopantelis.

**Supervision:** Evangelos Kontopantelis.

**Validation:** Evangelos Kontopantelis.

**Visualization:** Rathi Ravindrarajah, Evangelos Kontopantelis.

**Writing – original draft:** Rathi Ravindrarajah.

**Writing – review & editing:** Matt Sutton, David Reeves, Sarah Cotterill, Emma Mcmanus, Rachel Meacock, William Whittaker, Claudia Soiland-Reyes, Simon Heller, Peter Bower, Evangelos Kontopantelis.

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
