## [Editor Report · Decision Letter 0]

12 Apr 2022

Dear Dr Kontopantelis, 

Thank you for submitting your manuscript entitled "Effectiveness of the NHS Diabetes Prevention Programme at reducing conversion from non-diabetic hyperglycaemia to type-2 diabetes mellitus in England: a matched cohort analysis using electronic health records" for consideration by PLOS Medicine.

Your manuscript has now been evaluated by the PLOS Medicine editorial staff and I am writing to let you know that we would like to send your submission out for external peer review.

Please re-submit your manuscript within two working days, i.e. by Apr 14 2022 11:59PM.

Kind regards,

Beryne Odeny

PLOS Medicine

---

## [Decision Letter · Decision Letter 1]

2 Jun 2022

Dear Dr. Kontopantelis,

Thank you very much for submitting your manuscript "Effectiveness of the NHS Diabetes Prevention Programme at reducing conversion from non-diabetic hyperglycaemia to type-2 diabetes mellitus in England: a matched cohort analysis using electronic health records" (PMEDICINE-D-22-01190R1) for consideration at PLOS Medicine. 

Your paper was evaluated by a senior editor and discussed among all the editors here. It was also sent to independent reviewers, including a statistical reviewer. The reviews are appended at the bottom of this email and any accompanying reviewer attachments can be seen via the link below:

[LINK]

In light of these reviews, I am afraid that we will not be able to accept the manuscript for publication in the journal in its current form, but we would like to consider a revised version that addresses the reviewers' and editors' comments. Obviously we cannot make any decision about publication until we have seen the revised manuscript and your response, and we plan to seek re-review by one or more of the reviewers. 

We expect to receive your revised manuscript by Jun 23 2022 11:59PM. Please email us (plosmedicine@plos.org) if you have any questions or concerns.

We look forward to receiving your revised manuscript. 

Sincerely,

Beryne Odeny, 

PLOS Medicine

plosmedicine.org

1) Please include line numbers in the next draft.

2) Please revise your title according to PLOS Medicine's style. Your title must be nondeclarative and not a question. It should begin with main concept if possible. Please place the study design only in the subtitle (i.e., after a colon). For example, “Referral to the NHS Diabetes Prevention Programme and conversion from non-diabetic hyperglycemia to type-2 Diabetes Mellitus in England: a matched cohort analysis” 

3) The data availability statement needs revision. Given the use of CPRD data please consider the following statement or similar: ‘Requests for access to data from the study should be addressed to cprdenquiries@mhra.gov.uk. All proposals requesting data access will require approval from CPRD before data release.’

4) Abstract:

a) Please combine the Methods and Findings into one section “Methods and Findings”

b) Please include the important dependent variables that are adjusted for in the analyses.

c) Please ensure that all numbers presented in the abstract are present and identical to numbers presented in the main manuscript text.

d) Please quantify the main results (please present both 95% CIs and p values).

e) In the last sentence of the Abstract Methods and Findings section, please describe the main limitation(s) of the study's methodology.

5) Please delete the “Research in Context” section

7) Did your study have a prospective protocol or analysis plan? Please state this (either way) early in the Methods section. 

8) For additional context, please describe how the NDPP works and how the referral system is structured.

9) Your study is observational and therefore causality cannot be inferred. Please remove language that implies causality, such as “impact,” “effect” or “reducing the overall numbers.” Refer to associations instead.

10) Given the limitations of the study design, in the Introduction, please remove descriptions such as “rigorous estimate of ….”

11) The terms gender and sex are not interchangeable (as discussed in http://www.who.int/gender/whatisgender/en/ ); please use the appropriate term.

12) Please ensure that the study is reported according to the STROBE and include the completed STROBE checklist as Supporting Information. Please add the following statement, or similar, to the Methods: "This study is reported as per the Strengthening the Reporting of Observational Studies in Epidemiology (STROBE) guideline (S1 Checklist)."

13) Did you adjust for ethnicity and deprivation status based on location or patients’ addresses?

14) Please move “Ethical approval” to the Methods section

15) Results:

a)In the results, please ensure p<0.0001 is revised to p<0.001

b)Please quantify the main results with both 95% CIs and p values

16) Please define all abbreviations in Tables and Figures e.g., NDH, BMI, SD, T2DM

17) Please define S(t) on the y-axis of Figure 2

18) For the survival curve in Figure 2, please show the numbers at risk at regular intervals below the x axis.

19) In Figure 2, please show the axis beginning at zero. If this is not possible, please show a break in the axis.

20) Please remove subheadings in the Discussion section, and present and organize the Discussion as follows: a short, clear summary of the article's findings; what the study adds to existing research and where and why the results may differ from previous research; strengths and limitations of the study; implications and next steps for research, clinical practice, and/or public policy; one-paragraph conclusion.

21) Please remove the ‘Funding,” “Data sharing,” and “Completing interests” statements from the end of the main text. In the event of publication, this information will be published as metadata based on your responses to the submission form.

22) References:

a) Please select the PLOS Medicine reference style in your citation manager. In-text reference call outs should be presented as follows noting the absence of spaces within the square brackets, e.g., "... services [1,2]."

b) References should have six names before et al. 

c) Please ensure that journal name abbreviations consistently match those found in the National Center for Biotechnology Information (NCBI) databases. https://journals.plos.org/plosmedicine/s/submission-guidelines#loc-references. 

Comments from the reviewers:

Reviewer #1: The NHS Diabetes Prevention Programme (NDPP) is a behaviour-change programme for adults who are at risk of developing type 2 diabetes based on laboratory test results showing impaired fasting glucose or having an impaired glucose tolerance test. The objective of this study is to determine whether the NDPP referral decreases the risk of developing type 2 diabetes among individuals with non-diabetic hyperglycemia (NDH). This study uses the Clinical Practice Research Datalink to identify individuals with NDH referred to the NDPP matched to individuals with NDH who were not referred to the NDPP. The study findings showed that individuals with NDH referred to the NDPP had a lower risk of developing type 2 diabetes compared to individuals who were not referred to the NDPP (HR: 0.80; 95% CI: 0.73-0.87). This is an interesting study showing that the NDPP may have potential benefits in reducing the risk of developing type 2 diabetes among individuals who have pre-type 2 diabetes. I have some comments listed below:

Minor comments

1) In the Introduction section, the last statement states that the study assesses the impact of DPP in reducing the conversion of NDH to diabetes (incidence)-diabetes should be specified as type 2 diabetes (T2DM).

2) In the Introduction section, it would be interesting to provide information on the number of people referred to the NDPP, how many complete the program in general. Furthermore, in the discussion, it states that the NDPP is effective in reducing weight and HbA1c, can the authors include the magnitude of the effects? What is the drop out rate?

3) In the Methods section, the cohort defined NDH using Read code for Glucose Tolerance Test abnormal-wouldn't this include people with T2DM? Was this a typo?

4) In Figure 2, please define the abbreviation "stpm2"

Major comments

1) The results showed that individuals treated with metformin had a higher risk of developing T2DM. Is there concern of misclassification whereby individuals with metformin prescriptions likely have a T2DM diagnosis? Should individuals with prescriptions for anti-diabetic agents be excluded from entering the cohort to avoid including those with T2DM? Similarly, did the study investigators considered individuals treated with weight loss agents such as GLP-1RA?

2) In relation to comment #1, should diagnosis of T2DM be also defined as prescription of an antidiabetic agent in addition to the use of Read codes?

3) I am concerned that the comparison group consisted of individuals from a non referring practice. Is the care provided by these centers different compared to the care provided by centers that refer patients to the NDPP? 

4) Socioeconomic status (SES) information was not included in the complete case analysis which is a limitation since it was not available for individuals registered in non-English practices (which many were non referring centers). SES is a significant confounder that was not accounted for (i.e. people with lower SES likely have a higher risk of developing T2DM and may be less likely to participate in the NDPP). 

5) Given that people who participate in the NDPP may have healthier behaviors compared to people who do not participate, can the analyses adjust for proxies of healthy behavior such as physician visits/year, or vaccination uptake?

Reviewer #2: Thanks for the opportunity to review your manuscript. My role is as a statistical reviewer, so my review focuses on the study design, data, and analysis presented in the manuscript. I have put general comments first, and followed these with queries relevant to a specific section of the manuscript (with a page/paragraph reference).

The study used an interesting matching approach - using the staggered nature of the program introduction, they matched (stratified by available follow-up time) practices that begin referrals at a particular time-point to a practice that hadn't begun referrals (based on what I think is number of patient identified with NDH), and then within this practice-matched pool, exactly matched patients from one practice to the other (based on age (3 year distance), sex, and date of diagnosis (1 year window). This has the strength of making the unexposed patients contemporary with the exposed (those referred to the DPP) - this is overall a good approach to matching and I'm impressed they were able to do the exact matching (this never seems to work for me). The resulting selected cohort appears to be matched pretty well for most variables. A parametric survival model with shared frailty (practice) is used for the main analysis - I had a few queries about this but on the whole is a good approach given that it sounds like the provision of referrals sounds like it was made available at a practice level. The results are presented in both relative (HR) and absolute terms which is helpful. 

Typically the 'case/control' terminology is used to describe outcomes (T2DM incidence/no incidence), rather than exposures (exposed/unexposed, referred to NDH/not referred to NDH). I would consider updating the terminology throughout the manuscript, using referred to NDH/not referred is probably the clearest way to convey the information. 

P5, Paragraph 2. Are there differences in the regions covered by the two databases? Are the two databases generalizable to the population served by NHS?

P5, Paragraph 5. When I went to look at clinicalcodes.org, it was mainly content about steroid use in baseball. I'd consider dropping the 'www' so that people don't go directly to that (incorrect) URL, and include the .man.ac.uk URL in the reference list. 

P6, Paragraph 1. For the study period where matching was used, what was the distribution of referrals like? I was interested in what proportion of the total sites was non-referring, how many made 2 or more referrals, and how many made 1-19. 

I was unclear what the variable 'NDH registers' meant, is this the number of patients identified as having NDH? 

For the propensity score matching of sites, was there evidence of common-support?

P6, Paragraph 2. Were the covariates included in the form seen in appendix table 2/3? 

The missing-as-indicator approach relies on the data being 'missing-completely-at-random'. In particular, for unbiased estimates, that the presence of a missing value is not correlated with any other of the covariates. In the sensitivity analyses, a version of the analysis with multiple imputation is used and these general shrink the main estimate back to a null (HR=1) effect. This suggests that the missingness can't be considered MCAR. What was the rationale for the missing-as-indicator approach to be used for the main analysis given that the MI analysis is robust to data that is at least missing-at-random? 

P7, Paragraph 2. Was the flexible parametric survival model used in both the main analysis and for the analysis for average survival curves within groups? Where it was used, what was the form of the mdoel and how was it selected (e.g. number of knots)?

What was the rationale for using the bootstrapping? Was it a purely cluster-based bootstrap (i.e. each site was selected in each bootstrap sample with all patients), or was it a two-stage bootstrap (bootstrapping both site and patients within each site). 

P7, Paragraph 3. 

How was the number of imputations for the MI sensitivity analysis decided? 

It would also be nice for detail about the rationale for each of these sensitivity analyses (just a few sentences) in the appendix) as there is detail for some of these sensitivity analyses but not others. 

P9, Paragraph 6. The limitation of making a causal inference from observational data should be directly acknowledged here. 

Appendix Fig 1. This is a very clear figure to follow - is it possible to have a similar (separate) figure for the exclusion of sites? 

Appendix Table 2/3. Is it possible to get a column with standardised difference for each variable? Most of these look like a good match but it's hard to get a sense for variables with a high SD like follow-up from NDH diagnosis in Table 2. 

Reviewer #3: General Comment: This is a potentially important paper because, despite the hope and potential for wide-scale diabetes prevention programmes though behavioural support, the effectiveness of such approaches implemented in the real world have never been established. This paper is generally well written and the analytic design seems appropriate. But the paper falls far short in providing definitive conclusions due to lack of clarity and comprehensiveness in the analysis. My specific comments/suggestions are the following:

1. Study denominator selection: The abstract refers to "complete case analysis" and then we never hear of that again (unless I missed it) . Exactly what defined a complete case, why was this approach chosen, and what approaches were taken to understand the characteristics of the non-complete cases, and most importantly, what are the implications (on quantitative estimates) of leaving them out? These are issues that should be included in the methods, results, and particularly discussion section, where it is lacking. (At present the discussion spends a lot of time talking about nuances of the matching decisions, which is comparatively trivial.)

2. Effects of differential outcome measurement: There is no reporting of the potential for differential measurement that could affect detection biases. As this study is based on passive rather than active determination of incidence (ie., letting it happen and be reported through primary care, rather than standard measurements), there should be some assessment of whether intervention participants received more HbA1c measurements than non-participants. Table 1 actually suggests the opposite of that is the case, but that is % with values at all. Would also be helpful to see whether there is variable numbers of HbA1cs. This directly affects outcome ascertainment.

3. Putting this paper in context. This discussion, and to a lesser extent the introduction as well, does a poor job putting this paper in context of the other literature, which is enormous in other domains (RCTs; community translation trials, CE modelling, etc.). Neglecting more comment on this ends up underselling the 

4. Introduction section: The first 2 paragraph are a bit clunkily written. It goes from describing the problem, then to the evidence for prevention, then about what NDH is, then back to evidence for prevention. At the same time, the introduction does not provide the overall evidence context very well. There is far more that precedes this paper and makes it important than the handful of RCTs that are referenced. Suggest better focus and a re-write of the introduction.

5. NDH Case selection: Page 5, 3rd paragraph: Says this group explored how well NDH codes identify NDH but then stops there and doesn't tell us what they found, and if they are any good. In most systems, read codes or ICD coding is horribly non-sensitive and probably biased towards selecting high risk participants. Did that happen here? 

6. Matching process and selection: The matching approach seems solid, and in general, doing this at practice rather than individual level seems a strength; however, wo questions come up:

 a. Was measurement frequency (ie., A1c) considered as a variable in the matching process to limit the potential selection or detection bias?

 b. Does the use of practice based matching mean that if a given practice is more involved in NDPP referral, that they are also differentially involved in detection?

7. Analytic approach: following form comment 1 above, how is complete case analysis defined here, who exactly is included and excluded. All of this deserves good description in methods, showing implications of this approach in the results, and discussing it in the discussion. 

8. Referral definitions: Page 5, paragraph 4 describes all the different referral codes and participation levels apparently. Given that this is the crucial exposure, it would be nice to see what these codes represent, how they are defined, and how many. It appears that ALL of these groups considered as "cases"? Please clarify.

9. Additional minor point; calling these "cases" is awkward in a prospective trial emulation because they are not really cases of disease, they're actually non-cases and participants, some of whom become cases; I would encourage different terminology).

10. Description of findings. Several analyses would improve our ability to interpret the findings of this study, including:

 a. Presentation of numbers by each level of referral group (completed, vs not, etc.).

 b. A table comparing characteristics of the non-NDH, NDH, referred without NDH, referred and completed, attended, etc. ie., who are these people?

 c. Presentation of incidence rates of conversion (events per 1000 per year) for appropriate external comparison.

 d. Showing comparison of rates across these groups, eligible vs referred vs attended, etc.. 

 e. For the subset with serial HbA1c measurements, show those trends comparing participants and matched controls.

11. Minor point - the association of obesity with conversion and diabetes risk is usually curvilinear. Why not show the risk ration with class II and class III obesity, given that there is likely ample sample size.

12. Discussion section: This section really falls short in doing 3 things: 1) Explaining the context of this paper in the broader literature, so that the reader knows how potentially important this study is; 2) Addressing the potential biases that could be at play in a quasi-experiment like this, and more importantly, explaining what the most likely implications are on findings; 3) Discussing what this study really means for practice and policy.

Reviewer #4: Very important study on the effectiveness of referral to the programme at reducing conversion of NDH to T2DM. I dont have sufficient expertise to comment on the methodology, but the overall rationale and discussion of results were well-elaborated.

[LINK]

---

## [Decision Letter · Decision Letter 2]

16 Sep 2022

Dear Dr. Kontopantelis,

Thank you very much for submitting your manuscript "Referral to the NHS Diabetes Prevention Programme and conversion from non-diabetic hyperglycaemia to type-2 diabetes mellitus in England: a matched cohort analysis" (PMEDICINE-D-22-01190R2) for consideration at PLOS Medicine. 

[LINK]

In light of these reviews, I am afraid that we will not be able to accept the manuscript for publication in the journal in its current form, but we would like to consider a revised version that addresses the reviewers' and editors' comments. Please pay special attention to reviewer #3's concerns raised in the first round of review. We see that their comments have not be adequately addressed. 

We expect to receive your revised manuscript by Sep 16 2022 11:59PM. Please email us (plosmedicine@plos.org) if you have any questions or concerns.

We look forward to receiving your revised manuscript. 

Sincerely,

Beryne Odeny, 

PLOS Medicine

plosmedicine.org

Requests from the academic editor:

I have reviewed the reviewer 3's comments and reply from the authors. However, I don't think the authors answered the initial questions raised by the Reviewer 3 satisfactorily, especially the selection bias due to coding and the participants' willingness to receive measurement, etc. This will impact the interpretation of the outcome of this study. Although the topic of this study is interesting and important, there do exist some limitations in the methodological part, and the authors didn't solve these problems in their 2nd round revision. 

The authors should improve the display of the whole manuscript to make it more reader-friendly.

Comments from the reviewers:

Reviewer #1: I have reviewed the revised manuscript and the responses to my comments. The authors have addressed all my comments and I do not have any further concerns. Thank you for involving me as a reviewer. 

Reviewer #2: 

Thanks for the revised manuscript and answers to my queries. Overall most of my original queries have been resolved in this revision. Just a few minor points below.

Generally you don't always get a perfect match of propensity scores, so the use of calipers is not itself problematic. Just judging from the summary statistics, there was a reasonable degree of common support. 

For appendix 2/3 the suggestion was for standardised difference (between exposure groups), there is a good description of this in Austin 2009 (https://doi.org/10.1002%2Fsim.3697). This is fairly straightforward addition and while I agree that the table is already complex, this would probably simplify some of that complexity to highlight that is very limited differences after the matching process.

For Figure 2 I would consider inverting the y-axis (so it's is cumulative incidence of conversion) and limiting the range as well to visualise the differences more easily. If possible, labelling each group directly on the figure is probably the best way to indicate which line is Normal or DPP. 

It looks like the ClinicalCodes repository still has some issues with the domain - could you just include these in the appendix? They are an essential part of the reproducibility of the study.

[LINK]

---

## [Decision Letter · Decision Letter 3]

1 Dec 2022

Dear Dr. Kontopantelis,

Thank you very much for submitting your revised manuscript "Referral to the NHS Diabetes Prevention Programme and conversion from non-diabetic hyperglycaemia to type-2 diabetes mellitus in England: a matched cohort analysis" (PMEDICINE-D-22-01190R3) for consideration at PLOS Medicine. 

Your paper was evaluated by an associate editor and discussed among all the editors here. It was also discussed with an academic editor with relevant expertise, and sent back to independent reviewers, including a statistical reviewer. The reviews are appended at the bottom of this email alongside editorial comments.

In light of these comments, I am afraid that we will not be able to accept the manuscript for publication in the journal in its current form, but we would like to consider a revised version that addresses the reviewers' and editors' comments. We cannot make any decision about publication until we have seen the revised manuscript and your response. 

We hope to receive your revised manuscript by Dec 15 2022 11:59PM. Please email us (plosmedicine@plos.org) if you have any questions or concerns.

We look forward to receiving your revised manuscript. 

Sincerely,

Callam Davidson, 

PLOS Medicine

plosmedicine.org

After consulting the academic editor, the editorial team are concerned that some of the original issues raised by Reviewer #3 (on the R1 version) remain unresolved and require resolution before we can consider publication: 

* Relating to Reviewer #3, Comment 2 (R1 Version): It is the academic editor's understanding that, in Table 1, the rates with HbA1c values were much lower in the "referred to NDPP" group than in the "not referred to NDPP" group. The academic editor notes that this is quite strange and is curious to understand why. As reviewer #3 and the academic editor note, this observation is the opposite of what one would expect, and led reviewer #3 to raise concerns regarding detection bias. Although the academic editor acknowledges that the authors provided some explanation in their response (R2 version, e.g., more proactive health management by referred cases leading to earlier T2DM diagnosis), they do not find this to be sufficiently convincing. If this was the case, those referred to NDPP should pay more attention to glucose monitoring. Please provide further comment on this point.

* Relating to Reviewer #3, Comments 3 and 4: The academic editor feels that author responses to these comments (R2 version) did not address the issues raised and suggests that the authors reorganize the introduction and discussion sections to ensure clarity and to provide important context to aid reader comprehension. Please address past research and explain the need for and potential importance of your study. Indicate whether your study is novel and how you determined that. Please present and organize the Discussion as follows: a short, clear summary of the article's findings; what the study adds to existing research and where and why the results may differ from previous research; strengths and limitations of the study; implications and next steps for research, clinical practice, and/or public policy; one-paragraph conclusion.

Please update the line numbering to be continuous throughout the manuscript (rather than restarting on each new page).

Your study is observational and therefore causality cannot be inferred. Please remove language that implies causality, such as ‘effect/effectiveness’ throughout. Refer to associations instead.

Abstract Methods and Findings:

* Please include the study design (cohort study), population and setting (patients attending primary care in England), and main outcome measures.

* In the last sentence of the Abstract Methods and Findings section, please describe the main limitations of the study's methodology.

Please update the Abstract heading ‘Interpretations’ to ‘Conclusions’.

Please remove the ‘Funding’ section from the Abstract.

Thank you for providing an Author Summary. Under the question ‘What do these findings mean’, authors should reflect on the new knowledge generated by the research and the implications for practice, research, policy, or public health. Authors should also consider how the interpretation of the study’s findings may be affected by the study limitations. The current bullet points are more aligned with the question above (What did the researchers do and find?).

We request 2-3 single sentence bullet points per question in the Author Summary. The aim is to make your findings accessible to a wide audience that includes both scientists and non-scientists, so please just concentrate on getting across the key points. See https://journals.plos.org/plosmedicine/s/revising-your-manuscript for further guidance. 

Please relocate citations such that they precede punctuation. 

Thank you for including a STROBE checklist. Please update the checklist to use Section headings and paragraph numbers (rather than page numbers and paragraph numbers) as the page numbers will change in the event of publication.

Please ensure you cite your STROBE checklist and other items in the Supporting Information using the following guidance: https://journals.plos.org/plosmedicine/s/supporting-information

Please provide the unadjusted comparisons as well as the adjusted comparisons in Table 2 and include covariates that were adjusted for in the legend.

Please ensure the flags and footnotes are correct in Figure 1 (the symbol ^ is used but does not appear to have a corresponding footnote). 

Comments from the reviewers:

Reviewer #2: Thanks for the revised manuscript and replies to the earlier questions. 

Thanks for the codes included as the appendix - if there's a way to directly link to the codes on the website that would be handy.

The reversed y-axis on a survival plot (cumulative incidence) is a practice recommended by Pocock and Altman (https://doi.org/10.1016/S0140-6736(02)08594-X), I believe here that it would allow for a more straightforward interpretation of the graph and allow you to directly label the y-axis (proportion converted to T2DM) rather than needing a title and displaying the function. This is not a crucial change and if you feel it should stay as is, then take this just as a recommendation. 

Similarly the standardised difference is recommended for matching studies, and makes it much easier for the reader to see that the difference between the groups is limited.

[LINK]

---

## [Editor Report · Decision Letter 4]

9 Jan 2023

Dear Dr. Kontopantelis,

Thank you very much for re-submitting your manuscript "Referral to the NHS Diabetes Prevention Programme and conversion from non-diabetic hyperglycaemia to type-2 diabetes mellitus in England: a matched cohort analysis" (PMEDICINE-D-22-01190R4) for review by PLOS Medicine.

I have discussed the paper with my colleagues and the academic editor. I am pleased to say that provided the remaining editorial and production issues are dealt with we are planning to accept the paper for publication in the journal.

The remaining issues that need to be addressed are listed at the end of this email. Please take these into account before resubmitting your manuscript:

We look forward to receiving the revised manuscript by Jan 16 2023 11:59PM.   

Sincerely,

Callam Davidson, 

Associate Editor 

PLOS Medicine

plosmedicine.org

Comments from the Academic Editor:

In the introduction part, it's better to introduce life-style intervention first and then introduce the medication intervention part. Because this manuscript focuses on the participants of prediabetes, life-style intervention is still the first and most important way.

Lines 397-398: Please use round brackets (external) and square brackets (internal). 

Line 163: Typo (full stop in place of comma).

The ‘Contributorship statement’ and ‘Funding’ sections can be removed from the main text (both are captured as metadata via the submission form and will be published alongside the article).

---

## [Editor Report · Decision Letter 5]

19 Jan 2023

Dear Dr Kontopantelis, 

On behalf of my colleagues and the Academic Editor, Professor Weiping Jia, I am pleased to inform you that we have agreed to publish your manuscript "Referral to the NHS Diabetes Prevention Programme and conversion from non-diabetic hyperglycaemia to type-2 diabetes mellitus in England: a matched cohort analysis" (PMEDICINE-D-22-01190R5) in PLOS Medicine.

PRESS

Sincerely, 

Callam Davidson 

Associate Editor 

PLOS Medicine